# Wild and domesticated animal abundance is associated with greater late-Holocene alpine plant diversity

Sandra Garcés-Pastor [1,2,3] ✉, Peter D. Heintzman [1,4,5], Scarlett Zetter [1], Youri Lammers [1], Nigel G. Yoccoz [6], Jean-Paul Theurillat [7,8], Christoph Schwörer[9], Andreas Tribsch[10], Kevin Walsh [11], Boris Vannière[9,12], Owen S. Wangensteen [2], Oliver Heiri [13], Eric Coissac[14], Sébastien Lavergne [14], Lieveke van Vugt[9], Fabian Rey[13], Charline Giguet-Covex[15], Gentile Francesco Ficetola [16], Dirk N. Karger [17], Loïc Pellissier [18], Robert Schabetsberger[19], Jean Nicolas Haas[20], Michael Strasser [21], Karin A. Koinig [22], Tomasz Goslar [23], Sönke Szidat [24], PhyloAlps Consortium*, Antony G. Brown[1], Willy Tinner[9] & Inger Greve Alsos [1]

In the face of human land use and climate dynamics, it is essential to know the key drivers of plant species diversity in montane regions. However, the relative roles of climate and ungulates in alpine ecosystem change is an open question. Neither observational data nor traditional palaeoecological data have the power to resolve this issue over decadal to centennial timescales, but sedimentary ancient DNA (sedaDNA) does. Here we record 603 plant taxa, as well as 5 wild, and 6 domesticated mammals from 14 lake sediment records over the last 14,000 years in the European Alps. Sheep were the first domesticated animals detected (at 5.8 ka), with cattle appearing at the early Bronze Age (4.2 ka) and goats arriving later (3.5 ka). While sheep had an impact similar to wild ungulates, cattle have been associated with increased plant diversity over the last 2 ka by promoting the diversity of forbs and graminoids. Modelling of the sedaDNA data revealed a significantly larger effect of cattle and wild ungulates than temperature on plant diversity. Our findings highlight the significant alteration of alpine vegetation and the entire ecosystem in the Alps by wild and domesticated herbivores. This study has immediate implications for the maintenance and management of high plant species diversity in the face of ongoing anthropogenic changes in the land use of montane regions.

Alpine regions are known for their high diversity, but changes in both climate and human land use pose a threat to their future[1,2]. The relationship between climate and biodiversity in the Alps is complex, as climate affects factors such as temperature and precipitation, which in turn shape plant distributions and ecosystem dynamics. Short-term studies suggest that careful management through grazing may mitigate climate-change induced loss of species[3–5], but little is known about long-term processes. During the first half of the Holocene (-11 to 6 ka), climate was the main driver of vegetation. Rising temperatures initiated a successional shift, characterised by an initial increase in

A full list of affiliations appears at the end of the paper. *A list of authors and their affiliations appears at the end of the paper.
✉e-mail: sgarcespastor@gmail.com

forbs, followed by a transition to shrubs and forests as warming progressed. Over the last 5 ka, human impact has had a greater impact on the biodiversity of the Alps than climate change. Identifying the most influential activities is crucial for theoretical and practical reasons, with palaeoecological studies suggesting that agriculture was the main driver of plant diversity[6]. However, traditional palaeoecological proxies, such as pollen, do not allow for the identification of specific human activities or the roles of specific wild or domesticated mammalian herbivores in mountain ecosystems. Herbivorous mammals can only be broadly traced through remains (bones/skins), or indirectly through coprophilous fungal spores[7], faecal biomarkers[8], or the occurrence of grazing-related flora[9], leaving a gap in our understanding of their specific impacts on alpine vegetation. Emerging sedimentary ancient DNA (sedaDNA) techniques allow us to distinguish between the influences of wild and domesticated herbivores on plant diversity and ecosystem processes. SedaDNA can also uncover a hidden diversity of taxa typically under-represented by pollen, such as graminoids, forbs and many insect-pollinated plants[10,11]. With the study of mammalian sedaDNA, some single lake studies suggest that the rise of local plant diversity is strongly related to millennia of pastoral activities[2,12–14]. This provides new insights into historical biodiversity patterns and the role of large mammalian herbivores in niche development and plant species composition. Furthermore, if sedaDNA findings differ from pollen studies, it will significantly impact management and conservation strategies.

Here, we show plant diversity and mammalian distributions across the Alps during the Holocene, employing a lake sedaDNA dataset of 704 sediment samples. We apply sedaDNA analysis to reveal how climate, along with wild and domesticated animals, has shaped plant diversity in different areas of the Alps. We also investigate the impact of large herbivores and climate on the diversity and abundance of alpine species, specifically forbs and graminoids, shedding light on the effects of grazing behaviour on plant diversity. Our results demonstrate that sheep grazing exerts minimal pressure on vegetation, while cattle grazing increases plant species diversity and maintains open habitats. Wild ungulates also contribute to species diversity, forming diverse alpine grasslands. We find that temperature is negatively associated with total plant diversity, while precipitation plays a minor role. However, herbivores help mitigate the effects of rising temperatures, underscoring their importance in ecosystem management. Since the Bronze Age (4.2 ka), grazing by cattle, other domesticates, and wild animals has overridden the smaller effects of climate change. This reinforces the necessity of grazing and wildlife management to sustain alpine biodiversity in the face of ongoing climate change.

## Results and discussion
### A regional-scale plant and mammal sedaDNA dataset
We collected 704 samples from 14 alpine lake sediment records for sedaDNA analysis (1450–2529 m a.s.l.) (Fig. 1 and Supplementary Table 1). To place the samples in a chronological context, we radiocarbon dated a total of 178 plant macrofossil remains, with the resulting ages presented in thousands of calibrated years before present (ka) (see Supplementary Data 1, 2 and Supplementary Fig. 1). We generated plant and animal sedaDNA metabarcoding data by sequencing either the P6-loop region of the plant chloroplast trnL (UAA) intron or a section of the mammalian mitochondrial 16S locus (see Methods). Our bioinformatic pipeline resulted in the identification of 603 plant taxa with 43% identified to species level (Supplementary Data 3). We next grouped a total of 408 plant taxa into ecological categories related to light requirements (349), grazing tolerance (188) and altitudinal vegetation belts (285). For mammals, we obtained 121 unique animal sequences (Supplementary Data 4, 5). After removing potential contaminants, as well as non-

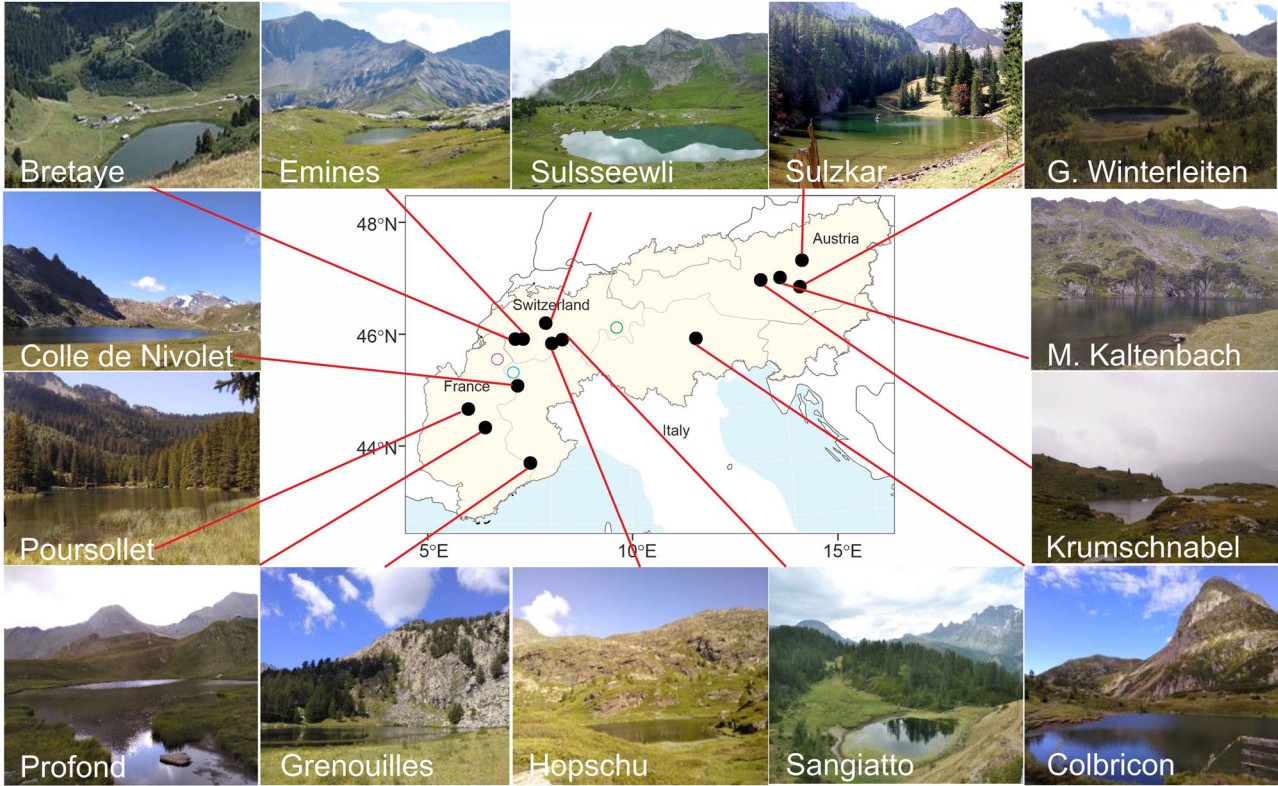

**Fig. 1 | Map of the Alps with the locations of the 14 lakes.** G. Grosser, M. Mittlerer. The circles correspond to lakes Verney (blue), Alterne (purple) and Chamfèr (green) ([18,38]). Photo credits: Bretaye, Emines and Sangiatto: Christoph Schwörer; Sulzkar: Erich Weigand, Sulsseewli: Inger G. Alsos; all others, Sandra Garcés-Pastor. Source Data can be found in Supplementary Data 11. Map is drawn from data from Natural Earth.

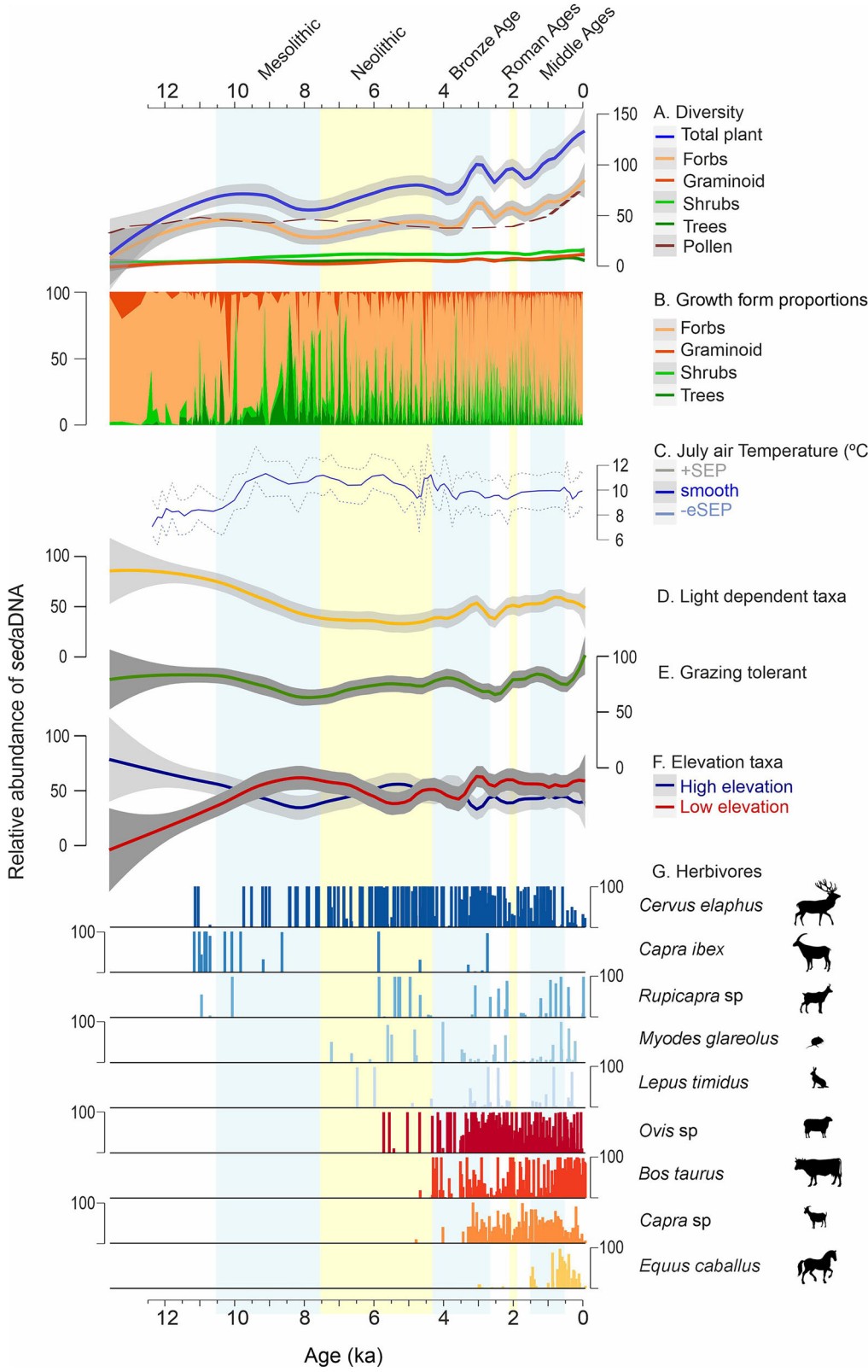

mammals, we retained 32 sequence variants representing six domesticated mammals (cattle: *Bos taurus*, sheep: *Ovis aries*, goat: *Capra hircus*, horse: *Equus caballus*, donkey: *Equus asinus*, camel: *Camelus* sp., Supplementary Data 6), and five sequences representing three large wild herbivores (ibex: *Capra ibex*, red deer: *Cervus elaphus*, chamois: *Rupicapra rupicapra*) and two small mammals (bank vole: *Myodes glareolus*, mountain hare: *Lepus timidus*).

**The impact of herbivorous mammals on vegetation dynamics**

The climate-driven changes in vegetation at the onset of the Holocene triggered a shift in ecosystem dynamics, leading to changes in wildlife abundance. During the Early Holocene, the study sites were generally characterised by open habitats dominated by herbaceous forbs from 11.7 to 9 ka (Fig. 2A, B and Supplementary Fig. 6). These consisted of light-dependent and grazing-tolerant plant taxa, along with high-

**Fig. 2 | Change in animal and plant diversity patterns in the Alps over the last 12 ka years. A** Hill N0 (diversity) of total plants, forbs, graminoids, shrubs and trees; Pollen richness from ref. 6, **B** *Sed*aDNA relative abundance index (RAI) percentages of vegetative growth forms, **C** Chironomid-reconstructed temperature from ref. 2 (dotted lines represent unsmoothed values of sample-specific estimated standard errors of prediction, +SEP and −SEP; the blue line shows the record smoothed using a three-sample running average), **D** light-dependent plant taxa (RAI), **E** grazing-tolerant taxa (RAI), **F** elevation taxa (RAI) and **G** herbivores *sed*aDNA: red

deer (Cervus elaphus), ibex (Capra ibex), chamois (*Rupicapra* sp), bank vole (*Myodes glareolus*), mountain hare (*Lepus timidus*), sheep (*Ovis* sp), cattle (*Bos taurus*), goat (*Capra* sp) and horse (*Equus caballus*). Vertical light yellow and light blue bars correspond to the Mesolithic, Neolithic, Bronze Age, Roman Age, and Middle Ages. All values except diversity and temperature are presented as RAI scaled according to the plotted groups. Source Data can be found in Supplementary Data 11. Silhouettes come from Phylopic https://www.phylopic.org/.

elevation species indicative of colder alpine and subalpine environments (Fig. 2C, E, F and Supplementary Figs. 7, 8). Red deer is recorded around 11.0 ka and then disappears again. Chamois is scattered, whereas ibex, known for their preference for open alpine habitats, has the highest rate of detection in this period. Between 8.5 and 6.0 ka, low-elevation plant taxa, along with trees and shrubs, associated with warmer environments become more abundant and both ibex and chamois are not detected. Red deer, a species that inhabits transitional zones between wooded or shrubby areas and open spaces, becomes more widespread (Fig. 2G). While one should be cautious when inferring taxon absence from *sed*aDNA data, this may indicate a bottom-up effect of vegetation on wild mammals inhabiting open habitats, providing a competitive advantage to red deer[15,16]. Following this warmer period, forbs, grazing tolerance, and light indicator taxa resurge at the expense of trees and shrubs (Fig. 2B–D). This rise in open landscape coincides with a reappearance of chamois and ibex from 6 ka onwards (Fig. 2G), aligning with the grazing behaviour typical of these animals[17]. Red deer stands out as the most readily-detected species, persisting over this period (Fig. 2G and Supplementary Figs. 9, 11).

Based on our *sed*aDNA data, sheep, cattle, and goats have been present in the Alps since the later Neolithic (5.8–4 ka), with a notable increase in cattle around 2 ka. Sheep is the first domesticated mammal that appears in the region, emerging in Colle del Nivolet (Western Alps) by 5.8 ka (Fig. 2G and Supplementary Fig. 10). During the later Neolithic (5-4 ka), sheep, goats, and cattle were introduced across the Western Alps (Supplementary Fig. 10). By the Bronze Age (4–3 ka), sheep and cattle appeared in the Central and Eastern Alps, coinciding with the first evidence of horses at 3.1 ka and a marked increase in grazing-tolerant plant taxa. This West-East trend aligns with the detection of sheep *sed*aDNA in lakes Verney and Anterne at around 3.6 ka in the Western Alps[18]. Over the last 2 ka, cattle and sheep were detected in all 14 of our lake sediment records (Supplementary Fig. 10). Goat occurrences varied greatly between lakes, with some showing simultaneous appearance with sheep while others lacked goats entirely. In our records, donkeys are exclusively present in Grenouilles and Sangiatto (Western Alps), whilst camels are detected in one sample each from Grenouilles (0.7 ka) and Profond (0.4 ka) in the Western Alps. We highlight that the use of camels in the Alps during the Middle Ages has been previously documented in the Grandes Chroniques de France[19]. Our data challenge earlier interpretations that claim domesticated livestock were in the Central Alps around 7 ka, based on indirect evidence such as coprophilous fungi, pollen pasture indicators, and archaeological evidence[20]. Before 5.8 ka, the only herbivores detected were red deer, which could explain the increase in coprophilous fungi and pollen pasture indicators in the region[21]. Our findings are supported by studies that register the human use of fire in alpine areas during the Late Neolithic/Early Bronze Age (~4.2 ka)[2,22,23], and they also expand our understanding by identifying the specific domesticated species involved. Thus, our data suggest that the arrival of domesticated mammals was later than inferred based on pollen and fungal spore data, and that increased grazing by wild mammals around 7 ka caused vegetation changes at that time.

### Ecosystem engineering by large and small herbivores
To explore the relative influence of climate (temperature and precipitation) and herbivore abundance (wild and domesticated) on plant

diversity, we employed Generalised Additive Mixed Models (GAMM) (Table 1; see Methods). Wild herbivores such as deer, chamois, and ibex show a significant positive correlation with the diversity of forbs and graminoids (Fig. 3B, C and Table 1). Wild herbivores may have shaped the ecosystem by grazing and trampling, thereby creating different patches of vegetation that enhanced sunlight penetration to the ground and reduced competitive plant growth. Trampling facilitated litter incorporation into the soil, creating openings for new plant growth and, in turn, increasing plant diversity[24]. Moreover, wild herbivores probably facilitated seed dispersal, while their urination and defecation increased available nutrients and accelerated nutrient cycling rates[25,26]. These results align with modern studies about the impact of deer, chamois, and ibex on the landscape[27–30]. Notably, the bank vole (20– 40 g) exhibited a negative correlation with forbs and graminoids (Table 1), likely due to its habitat preference for deciduous forests[31].

The introduction of medium-sized domesticated mammals, such as sheep (45–160 kg), during the Bronze Age had some effect on forb and shrub diversity, as well as a negative correlation with the abundance of trees and grazing-related taxa (Table 1). These effects mirror those observed with deer and chamois, except for the negative correlation with grazing-related taxa. In contrast, goats did not show a significant relationship, which might be related to their browsing behaviour. Conversely, horses did exert some influence on graminoid diversity. This pattern, suggesting that sheep, goats, and horses did not exert greater pressure than wild herbivores, has not been previously explored in palaeoecological studies. In contrast, the introduction of cattle (300–1400 kg) strongly influenced the diversity of forbs and graminoids, as well as the abundance of forbs, light-dependent, and grazing indicator taxa. Despite not matching the mass of their modern counterparts and probably living in smaller herds[32], cattle exacerbated the pressure on the vegetation exerted by wild animals. Consequently, the grazing pressure of these large herbivores and the forest clearance by humans prompted a shift in the ecosystem towards more open areas and diverse grasslands. The extent of this impact might have resulted from the cumulative influence of wild and domesticated animals over the past millennia. This is in line with contemporary studies that provide strong evidence of the positive effects of megafauna on plant diversity[33]. Our findings show both bottom-up and top-down interactions between plant diversity and both wild and domesticated animals. We reveal the influence of wild herbivores on vegetation, especially deer, chamois, and ibex, highlighting the significant role these animals play in driving plant diversity. This interaction has been well studied in contemporary systems, showing a rise in plant diversity with higher red deer density[29,34], but has proven challenging to discern with traditional palaeoecological techniques. Our findings support the hypothesis that livestock farming and wild mammals are associated with the growth of floristic diversity since the onset of the Bronze Age.

### Cattle enhance total plant diversity in the Alps
Our *sed*aDNA data show a continuous increase in plant diversity throughout the Holocene, with notable peaks around 3.2 ka and 2 ka, followed by a step increase (Fig. 2A and Supplementary Fig. 4). Most identified taxa are forbs, underscoring their ecological significance and potential sensitivity to environmental changes (Supplementary

**Table 1 | Regression coefficients of the linear mixed model applied to total plant diversity (Hill NO), diversity within growth forms (Hill NO) and relative abundance of growth forms, light, and grazing-related taxa for the Holocene**

| | Temperature | Precipitation | Deer | Chamois | Ibex | Cattle | Sheep | Horse | Bank vole | Mountain hare |
|---|---|---|---|---|---|---|---|---|---|---|
| Total plant diversity | –5.99 | 0.03 | 6.78 | 12.34 | 10.48 | 26.3 | 8.89 | | –7.37 | |
| Diversity within growth forms | | | | | | | | | | |
| Forbs | | –0.04 | 4.21 | 9.51 | 8.65 | 18.98 | 6.47 | | –5.19 | |
| Graminoids | –1.15 | | | 1.53 | 1.46 | 2.99 | | 0.87 | –1.35 | |
| Shrubs | –0.76 | 0.01 | 0.75 | 1.03 | | 2.02 | 1.21 | | –0.75 | |
| Trees | –0.69 | 0 | 0.48 | | | 0.45 | | | | |
| Abundances within growth forms | | | | | | | | | | |
| Forbs | 0.35 | –0.04 | | 5.75 | 4.7 | 6.38 | | | | |
| Graminoids | –1.56 | | | | | | | | | 2.49 |
| Shrubs | 2.25 | 0.02 | | | –4.23 | –3.34 | 2.83 | | | |
| Trees | 1.98 | 0.02 | –1.68 | –3.13 | | –3.19 | –2.06 | | | |
| Light indicator taxa | | –0.1 | | 9.72 | 8.21 | 8.74 | | | | |
| Grazing indicator taxa | | –0.02 | | | | 4.9 | –4.57 | | | |

GAMM results and figures in the Supplementary Code 1.

Fig. 5). The rise in plant *sed*aDNA diversity at 3.2 ka coincides with the co-occurrence of most domesticated (sheep, cattle, goats, horses) in our regional-scale record. This aligns with elevated erosion rates documented towards the end of the Bronze Age in the western European Alps[13]. A pivotal point arises when cattle consistently appeared in all lake records, during the Iron Age and into the Roman Period (2 ka), triggering an accelerated increase in diversity at the alpine regional level. While the latter rise is also observed in pollen data, the overall fluctuations in total plant diversity differ from richness graphs based on pollen[6,35–37], offering a more detailed trend (Fig. 2A). *Sed*aDNA unveils a higher number of plant taxa, especially forbs, which are associated with most peaks.

Our models suggest that cattle have the strongest positive correlation with total plant diversity across our sites, with weaker associations with temperature, deer, chamois, ibex, bank vole, and sheep (Fig. 3A and Table 1). Cattle have strong positive correlations with forb diversity and abundance, followed by abundances of light-indicator and grazing-related taxa. Furthermore, cattle also have a negative correlation with shrub and tree abundance, more typical of forested areas. These findings support and regionalise previous studies conducted at Sulsseewli[2] and Grosser Winterleitensee[38], two of the lakes included here, indicating that the establishment of transhumance during the Bronze Age, and especially the transition to cattle farming in the Roman period, was significantly associated with plant diversity. Our empirical data across the Alps highlight that cattle are strongly associated with plant diversity over the last 2 ka. These results are consistent with contemporary studies linking land abandonment to a notable decline in plant diversity[5,39]. Understanding the past role of herbivores offers fundamental insights into the history, dynamics, and conservation of present-day plant communities. In summary, our study indicates that size and grazing behaviour matter when introducing domesticated herbivores.

This study, using *sed*aDNA from 14 alpine lake records, unravels the complex interplay of climate, wild and domesticated herbivores, and human activities in shaping high-altitude vegetation over the Holocene. We reveal that, while sheep grazing mirrors pressure from wild mammals and has a minimal impact on vegetation, cattle impose higher grazing pressure, leading to increased plant species diversity and the maintenance of open habitats. However, the presence of wild ungulates has also enhanced species diversity, and has effectively created species-rich alpine grasslands. Regarding climate, precipitation had a minor impact on total plant diversity, while temperature exhibited a negative correlation. However, to mitigate the effects of rising temperatures, the influence of herbivores becomes crucial. This research underscores the significance of incorporating large herbivores into ecosystem management and conservation strategies. Maintaining pastoral activities at the treeline and lower alpine levels can hinder successional processes accelerated by warming, transforming species-rich pastures into species-poor alpine heaths, green alder scrubs, or coniferous forests[4,40]. GAMM modelling of the *sed*aDNA reveals that in combination cattle, other domesticated and wild animals overrode the smaller effect of climate change since the Bronze Age (4.2 ka). This supports the criticality of grazing and wildlife management for maintaining subalpine and alpine ecosystem diversity in the face of ongoing climate change.

## Methods
### Study Sites and core sampling
The 14 lakes of this study cover almost all the countries that comprise the Alps: four from Austria, three from Italy, four from Switzerland, and three from France (Fig. 1 and Supplementary Table 1). All necessary permits were obtained for fieldwork, including access to lakes and sediment coring. Permission to access the lakes and retrieve sedimentary cores using a floating platform and a boat with an electric motor was granted for the following locations: Colle del Nivolet (Gran Paradiso National Park), Grenouilles (Parc national du Mercantour), Colbricon (Parco Naturale Paneveggio Pale di San Martino), Sangiatto, Hopschu, Sulsseewli, and Bretaye. For Lake Sulszkar, no specific permit was required for coring, as approval had already been granted by the National Park. Twelve of the fourteen sedimentary cores were obtained during the summer of 2018, and an additional core from Sulzkar was retrieved in autumn 2020. Lake Bretaye (Switzerland) was previously collected in 2017. The Swiss cores were taken with a modified Streif-Livingstone piston corer[41] with 1 m long and either a 5 or 8 cm diameter pipe. The other cores were taken with a Nesje corer[42] and an up to 5 m long and 10 cm diameter ABS polymer pipe. Lake Sulzkar was taken with a UWITEC platform and gravity corer system with a maximum tube length of 2 m and a diameter of 8.6 cm. Lake depth was measured using a single-beam echosounder (Echotest II Plastimo) and tape measure. Coring and retrieval of sediments were performed from a

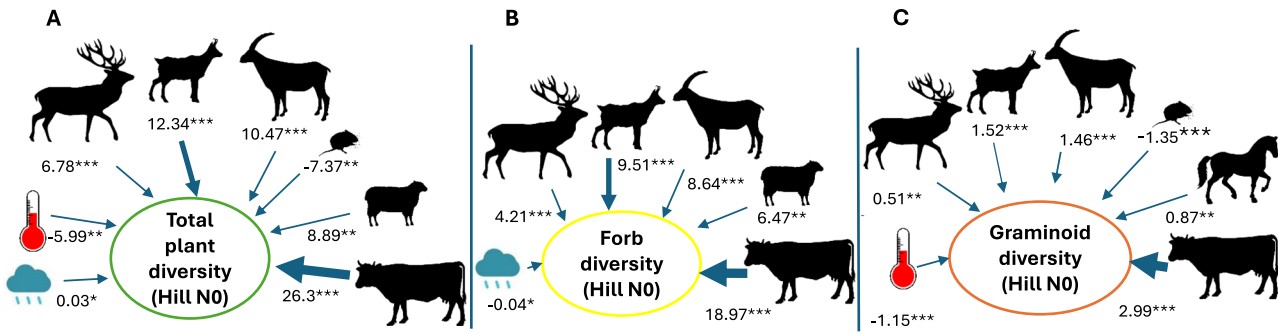

**Fig. 3 | Correlations between the diversity of plant groups and climate variables and mammal abundances.** Graphic outline of the GAMM depicting the significant correlations among temperature, precipitation, cattle, red deer, chamois, sheep, horse and bank vole with the diversity (Hill N0) of several plant groups **A** total plant, **B** forbs and **C** graminoids. Significance codes of *t*-value **** <0.0001; *** <0.001; ** <0.01. Source Data can be found in Supplementary Code 1. Silhouettes come from Phylopic https://www.phylopic.org/.

custom-made Zodiac boat-based floating platform. Sediment cores were cut into 1 m sections and immediately sealed to minimise the risk of modern DNA contamination. The Livingstone cores were extruded into half-pipes and immediately sealed at the facilities of the Institute of Plant Sciences & Oeschger Centre for Climate Change Research (University of Bern, Switzerland). Core sections were stored at 4 °C in the cold room at The Arctic University Museum of Norway (TMU, Norway) or, for the Lake Sulzkar core, at the University of Innsbruck (Austria) until they were opened by longitudinal splitting. After splitting, one half was used for *sed*aDNA subsampling, and the other half was retained as an archive.

### Construction of core chronologies

We collected 178 plant macrofossil remains for radiocarbon dating while sampling for *sed*aDNA. We obtained 10 to 20 macrofossils from each core, roughly according to the total sediment length (Supplementary Data 3). For each dated level, accelerator mass spectrometry (AMS) dating was typically performed on one or two well-preserved macrofossils. We sampled fewer macrofossils for Grenouilles, Emines and Bretaye because we could correlate our new dates with chronologies derived from previous palaeobotanical studies[43–45]. Macrofossils were dated using AMS at either the Poznań Radiocarbon Laboratory of the Adam Mickiewicz University, Poland ('Poz' accessions) or the Laboratory for the Analysis of Radiocarbon at the University of Bern ('BE' accessions) (Supplementary Data 1). All AMS radiocarbon dates were calibrated using the terrestrial IntCal20 curve[46] within age-depth models that were constructed using the Bayesian framework calibration software 'Bacon' v.2.3.4[47] in R v3.4.2 (R Core Team 2017).

We generated a composite core from multiple cores at the same site (Bretaye, Sulsseewli, Emines, Sangiatto, Hopschu, Sulzkar), by aligning them using visible stratigraphy and radiocarbon dates. The reported depths are derived from the composite cores and are measured from the water-sediment interface.

The Bayesian age-depth models for the 14 lakes span a period ranging from 12 to 0 ka (Supplementary Fig. 1). Most of the lakes exhibit continuous sedimentation, except for Krumschnabel, which experiences an instantaneous deposit between 60–95 cm (-0.69–0.74 ka), and Grenouilles and Sulzkar (with several few centimetres, to occasionally dm-scale instantaneously-deposited sedimentary layers). These instantaneous deposits were removed to build an 'event-free' age-depth model (e.g. refs. 48,49).

### Sedimentary ancient DNA data generation

Cores were subsampled for *sed*aDNA every 4-12 cm, with the interval dependent on total core length, by taking a 1 cm thick longitudinal subsample while avoiding the first 4–10 mm of exposed sediment[50].

Subsampling was performed in the ancient DNA lab at TMU using sterile tools, a full bodysuit, and gloves to ensure uncontaminated samples. For Sulzkar, subsampling was conducted in a cleaned teaching lab where no molecular biology work had been previously carried out at the Department of Ecology, University of Innsbruck in Austria (Supplementary Data 2). All *sed*aDNA extractions were conducted in a clean ancient DNA laboratory at TMU and followed the protocols of ref. 51. Several negative controls were used during the extraction and amplification process (Supplementary Table 2): controls C1 and C2 were related to sampling from the core, C3 and C4 to detect contamination during DNA extraction, and C5 was associated with potential contamination during the movement of DNA extracts from tubes to plates in preparation for metabarcoding PCR. At the same time, C6 served as a negative PCR control. C7 controls consist of DNA-free water added in the post-PCR laboratory and serve as our internal track of the amplicon contamination level in that room. The positive control (C8) used in the post-PCR lab consists of synthetic standards (see ref. 51), and their purpose is purely to check that the PCR has worked. As the sample PCR reactions are not exposed to the PCR-lab environment, controls C7 and C8 were removed during the bioinformatic processing and not considered in downstream analyses. DNA was extracted from 704 sediment samples and 105 sampling or extraction negative controls using a modified DNeasy PowerSoil kit (Qiagen 12855-100, Germany) protocol. DNA extracts and negative controls, along with 17 negative and 18 positive PCR controls, were amplified using uniquely dual-tagged universal generic primer sets[2] that amplify plants or mammals separately. The lookup dataset for all tag and sample combinations is available from Zenodo (https://doi.org/10.5281/zenodo.14283341). For plants, the primer set targets the *trn*L P6-loop region of the chloroplast UAA intron (gh primers, 10–143 bp[52]);. These primers were specifically designed by ref. 52 to target two highly conserved catalytic regions of the *trn*L (UAA) intron. Although primer-binding site mismatches are observed in some families such as Asteraceae (g: 0%, h: 19.8%), Caryophyllaceae (g: 0.9%, h: 6.8%) and Crassulaceae (g: 0%, h: 6.2%), the mismatch is ≤1% for other families[53]. This locus has been shown to provide the highest taxonomic resolution in ancient plant *sed*aDNA studies[50,54–56]. For mammals, a section of the mitochondrial 16S locus was amplified (MamP007 primers, 60–84 bp[57]). Reaction and cycling conditions for all PCRs followed[2]. PCR reactions were performed in 40 µL final volumes containing 1× Gold buffer, 1.6 U Ampli-TaqGold DNA Polymerase (Life Technologies, 10360545), 2.5 mM MgCl₂, 0.2 mM dNTPs (VWR, 733–1854), 0.2 µM of each primer, 160 ng/µL Bovine Serum Albumin (Fisher, 10829410), and 4 µL of DNA extract. For mammalian mitochondrial 16S PCRs, two modifications were applied: (1) forward and reverse primer concentrations were reduced to 0.1 µM each, and (2) forward and reverse blocking primers were added at 1 µM each. Cycling conditions included an initial enzyme activation at 95 °C for 10 min, followed by 45 cycles of denaturation at 95 °C for 30 s, annealing

at 50 °C for 30 s, and elongation at 72 °C for 1 min, with a final elongation step at 72 °C for 7 min. As recommended for *sed*aDNA metabarcoding studies, eight PCR replicates were carried out for each sample or control for both *trn*L and 16S[58]. Up to 384 PCR replicates were pooled and cleaned with a Qiagen MinElute PCR purification kit (Qiagen, 28006). Each pool was converted into a DNA library using a modified TruSeq PCR-free library kit (Illumina, 20015962) and unique dual-indexing (Illumina, 20022370)[51]. The libraries were quantified by qPCR using the Library Quantification Kit for Illumina sequencing platforms (KAPA Biosystems, 7960140001, Boston, USA), using a Prism 7500 Real-Time PCR System (Life Technologies, The Norwegian College of Fishery Science, the Arctic University of Norway UiT). Each library was sequenced on ~10% of a flow cell on the Illumina NextSeq-550 platform (2×150 bp, mid-output mode, Illumina, 20024905) at the UiT Genomics Support Centre in Tromsø, except for lakes Grosser Winterleiten (EG23) and Sulzkar (EG42) that were sequenced on a MiniSeq at TMU and a MiSeq at the Norwegian College of Fishery Science, respectively.

### Database construction and *sed*aDNA data analysis

The OBITools software package[59] was used for the bioinformatics pipeline, following the protocol and criteria defined by ref. 51. Briefly, paired-end reads were aligned using SeqPrep (https://github.com/jstjohn/SeqPrep/releases, v1.2). Merged reads were demultiplexed according to their 8 bp unique primer tags and identical sequences were collapsed. Singleton sequences and those shorter than 10 bp were removed, and putative artifactual sequences were identified and removed from the dataset[60].

The taxonomic assignment of the P6-loop reads was then carried out for the four different reference libraries following[2]: (1) PhyloAlps (http://phyloalps.org,2); (2) ArctBorBryo (regional arctic/boreal reference library compiled from refs. 53,61) and ref. 62; (3) PhyloNorway[55,63] and (4) the global reference library based on the EMBL rl143 database. For mammals, we generated a reference library from the EMBL rl143 database. The identified sequences were filtered in R using a custom script (available at https://github.com/Y-Lammers/MergeAndFilter). Only plant sequences with a 100% match to a reference sequence, represented by three reads in a PCR replicate, and with a minimum of 10 total reads and three replicates across the entire dataset were retained[64]. Furthermore, sequence variants that were the result of PCR or sequencing errors, such as homopolymer length variation, were merged with the source sequence (following ref. 51). Additionally, sequences that displayed a higher average frequency in PCR replicates of negative extraction or PCR controls than lake sediment samples were removed, as well as common laboratory contaminants (Supplementary Data 8). Final taxonomic assignments for plants were based on the collapsed assignments to the four reference databases, giving priority to the local PhyloAlps identifications. LULU software was used to identify sequences that resulted from sequencing errors such as homopolymer length variation[65]. After a secondary manual curation, all sequences were merged with the source sequence (Supplementary Data 3, 10 and Supplementary Table 3). Off-target amplifications from the g-h primer set (bryophytes and algae) are reported in Supplementary Data 3 and for the MamP007 primer set (birds, fish, and invertebrates) in Supplementary Data 4. The average length and GC content of our final sequence dataset were $47.02 \pm 11.63$ bp and $31 \pm 7.19\%$, respectively, which closely resemble the values from the total PhyloAlps P6-loop reference database ($50.83 \pm 13.71$ bp and $30.37 \pm 7.98$ %), thus confirming that the DNA is of reasonable quality.

The identified taxa were compared with the species in Flora Alpina[66] and Flora indicativa[67]. We removed 67 samples for plants with low metabarcoding data quality, which had technical quality (MTQ) scores <0.52 and/or analytical quality (MAQ) scores <0.25 (Supplementary Figs. 2, 3 and Supplementary Data 3). These MTQ and MAQ score thresholds were derived from the negative controls (excluding those from the PCR lab, Supplementary Data 3), which had maximum MTQ and MAQ scores of 0.50 and 0.24, respectively, following the protocol of ref. 51. To evaluate plant diversity across time, we measured Hill N0 numbers (commonly referred to as species richness) using the proportion of weighted PCR replicates (wtRep).

In the 16S dataset, sequences exhibiting a 95% or higher match were assigned final taxonomic classifications through careful examination of NCBI BLAST hits[68] and comparison with established general biogeographic distributions (full justifications are given in Supplementary Data 4). Data for 16S sequences assigned to the same taxon were collapsed. In this paper, we only focussed on mammals.

Some domesticated and wild taxa (cattle, sheep, goat, deer) can be sources of sporadic PCR contamination. We observed occasional contamination of cattle (1%), sheep (5%), and deer (3%) in our >100 negative controls (Supplementary Data 9). Consequently, we need to approach sporadic appearances with caution. Although we retained detections from all samples, we interpret sporadic occurrences of these taxa, defined as single, isolated PCR replicate detections, as possibly deriving from contamination. We note that these occur in otherwise low diversity samples that are most at risk of sporadic contamination detections (e.g. sheep, goat and cattle before 8 ka) (Fig. 3 and Supplementary Fig. 9). Thus, for this study, we excluded single occurrences of domestic cattle (*Bos taurus*), sheep (*Ovis aries*), and goat (*Capra hircus*) before 8 ka, such as those in Bretaye, Sulsseewli, Sulzskar, Colbricon, Emines and Poursollet (Supplementary Data 7).

### Ecological indicators from plant *sed*aDNA

We cross-referenced the haplotype-sharing species with ecological parameters from Flora Indicativa[67] related to open areas (light-taxa), grazing disturbance, and altitudinal vegetation belts. We extracted a total of 408 of our 603 identified plant taxa that were informative of open areas (349), grazing (188), and altitudinal vegetation belts (285). These *sed*aDNA results are expressed as the relative abundance index (RAI)[2], which integrates information from the relative proportion of reads and replicability of metabarcoding PCRs. The RAI is a proportional index for each taxon within a sample, calculated by multiplying the proportion of obtained reads by the proportion of weighted PCR replicates (wtRep). The latter serves as a barcode relative detectability measure to account for variations in the relative read counts across all retained barcodes within a sample[2,51]. These values are then normalised for plotting (scaled), dividing by the sum of all RAIs for all species plotted in the figure. Plots were made with R v4.4.0 using the *vegan*, *rioja* and *ggplot2* packages[69–71].

### Temperature and precipitation time series

We used the mean July air temperatures and total annual precipitation for the last 12 ka obtained from the CHELSA-TraCE21k v1.0 model[72,73]. CHELSA-TraCE21k provides 1 km resolution climate data for temperature and precipitation in 100-year time intervals for the last 21,000 years. Palaeo-orography at high spatial resolution and at each timestep is created by combining high-resolution topography and information on the present, and historical glacial cover with the interpolation of a dynamic ice sheet model (ICE6G) and a coupling to mean annual temperatures from CCSM3-TraCE21k. Based on the reconstructed palaeo-orography, the mean annual temperature is downscaled using the CHELSA V1.2 algorithm to a 1 km resolution. The time series was finally extracted for the geographic coordinates of the 14 lakes. The values corresponding to the *sed*aDNA samples were obtained using the nearest neighbour interpolation.

### Statistical analyses

We employed multiple (generalised) additive mixed models (GAMM)[74] to investigate past changes in plant diversity (Hill N0) in response to climate (temperature and precipitation), and wild (ibex, chamois, deer) and domesticated (goat, sheep, cattle) mammal abundances. We decomposed the variation in temperature and precipitation as within

and between lake components. More specifically, we used these variables as predictors of total plant diversity, the diversity and abundance within plant growth forms, and the prevalence of light and altitudinal indicator taxa. Lakes were used as random effects. We used a Gaussian distribution for the response (species diversity), as the variance of residuals was approximately constant. No evidence of temporal auto-correlation of the residuals was found. Analyses were performed using *mgcv* package in R (R Core Team 2018).

## Reporting summary

Further information on research design is available in the Nature Portfolio Reporting Summary linked to this article.

## Data availability

The lookup dataset for all tag and sample combinations generated in this study have been deposited at the Zenodo repository (https://doi.org/10.5281/zenodo.14283341). The full accession codes for projects are PRJEB85748 https://www.ebi.ac.uk/ena/browser/view/PRJEB85748 and PRJEB52290: https://www.ebi.ac.uk/ena/browser/view/PRJEB52290. The raw OBITools output data, both the P6-loop (all reference libraries) and 16S datasets, are available at the Zenodo repository (https://doi.org/10.5281/zenodo.15132382). The Source Data for Figs. 1–3 and Supplementary Figs. 1–11 generated in this study are provided in the Supplementary Data 11. For acces to any raw materials or core sediments from this study, please contact Dr. Inger Greve Alsos.

## Code availability

The R code used for the statistical analyses can be found in Supplementary Code 1. The identified sequences were filtered in R using a custom script (available at https://github.com/Y-Lammers/MergeAndFilter).

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

## Acknowledgements

We would like to acknowledge the field assistance provided by Kristin Heggland, Kelsey Lorberau, Willi Tanner, and Roland Kaiser. Special thanks to Giuliana from Rifugio Alpino Laghi di Colbricon for her support during fieldwork. We express our appreciation to Richard Bonet, Scientific Director of the Ecrins National Park, France, for his support and

assistance in coring Lake Profond. We also thank Bruno Bassano from Gran Paradiso National Park and Giuseppe Bogliani from the University of Pavia for their support and contributions to the research programme. We thank Magdalena Kaltenbrunner, Alexander Maringer, and Herbert Wölger from the National Park Gesäuse for their support and permission to core Lake Sulzkar. Richard Niederreiter and Martin Niederreiter (Uwitec, Mondsee, Austria) supported the fieldwork of Lake Sulzkar. Julia Rechenmacher, Brigitte Hechenblaickner, Moritz V. Ladurner and Werner Kofler assisted with age-depth modelling, sediment sampling, preparation of plant macrofossils for radiocarbon dating, and palaeoecological analyses. Special thanks to the Bergschaft Suls for coring permission and to César Morales-Molino for his assistance with the fieldwork at Swiss lakes. Marie K. F. Merkel, Iva Pitelkova, and Jennifer Alejandrina Carbonell Ellgutter provided laboratory support. Bioinformatic analyses were performed on resources provided by UNINETT Sigma2—the National Infrastructure for High-Performance Computing and Data Storage in Norway for bioinformatic analyses. Additionally, we acknowledge the GRICAD infrastructure (https://gricad.univ-grenoble-alpes.fr) for supporting computations related to the elaboration of PhyloAlps data. I.G.A. and Y.L. were supported by the European Research Council (ERC) under the European Union's Horizon 2020 research and innovation programme (grant agreement No 819192) and the ECOGEN project "Ecosystem change and species persistence over time: A genome-based approach," financed by Research Council of Norway grant 250963/F20, the latter also supported P.D.H., S.G.-P., and S.Z., most fieldwork, and other running costs. Open access funding was provided by UiT The Arctic University of Norway. S.G.-P. was also supported by the Beatriu de Pinós Programme (BP-2021-00131) and a fellowship from "la Caixa" Foundation (ID 100010434, fellowship code LCF/BQ/PI24/12040011). P.D.H. acknowledges support from the Knut and Alice Wallenberg Foundation (KAW 2021.0048 and KAW 2022.0033). Coring and stratigraphic analyses at Lake Sulzkar were funded through an Earth System Sciences grant from the Austrian Academy of Sciences: "Pulling the plug—Restoration of an Alpine lake". Coring at Lakes Sangiatto, Hopschu, Emines, Bretaye, and Sulsseewli was supported by the Oeschger Centre for Climate Change Research (OCCR) as part of the Graduate School of Climate Sciences at the University of Bern. The PhyloAlps reference database was built thanks to the following projects: the joint ANR-SNF project Origin-Alps (ANR-16-CE93-0004, SNF-310030L_170059), European Research Council under the European Community's Seventh Framework Programme FP7/2007-2013 grant agreement 281422 (TEEMBIO), and by the SNF grant 31003A_149508. The sequencing for the PhyloAlps reference database was performed within the framework of the PhyloAlps project, funded by France Génomique (ANR-10-INBS-09-08).

## Author contributions

I.G.A., N.Y., W.T. and L.P. designed the research, raised the funding and provided resources; S.L., P.A.C. and E.C. generated and curated the PhyloAlps database; I.G.A., P.D.H., S.G.-P., Y.L., A.G.B., C.S., L.V., F.R., W.T., B.V., A.T., C.G.-C., R.S. and K.W. did the fieldwork; S.G.-P. and S.Z. did the ancient DNA laboratory work with input from I.G.A. and P.D.H.; T.G. and S.S. performed radiocarbon dating; C.S. and L.V. built the Swiss and Sulzkar composite cores. C.S., S.G.-P., J.N.H., M.S. and S.Z. performed age-depth modelling; R.S. provided access to the core from Sulzkar and, together with J.N.H., M.S. and K.K., contributed stratigraphic analyses. D.N.K. and L.P. developed the temperature time series for each lake location; S.G.-P. verified and curated the plant barcode sequence taxonomic assignments with input from J.P.T. and A.T., who also verified their botanic origin; P.D.H. verified the mammal barcode sequence taxonomic assignments, together with S.Z., and performed the plant sedaDNA data quality analyses; O.W. and S.G.P. designed the bioinformatic pipeline to obtain the indicator sequences, which were verified by J.P.T.; S.G.-P., P.D.H., S.Z. and Y.L. performed bioinformatics and the quality control checks; S.G.-P. and N.Y. did the statistical analyses; S.G.-P., P.D.H. and S.Z. curated the data; W.T., I.G.A., S.L., C.S., J.P.T., A.G.B., G.F.F., F.R. and O.H. contributed with the palaeoecological and botanical interpretation; P.D.H., S.Z. and K.W. provided interpretation of the mammalian data; K.W. provided archaeological interpretation of the region. S.G.-P., S.Z., I.G.A. and O.H. wrote the first draft of the manuscript, of which J.P.T., P.D.H., O.H., A.G.B., W.T., C.S. and L.P. and the remaining co-authors commented on. All authors have reviewed and approved the final manuscript.

## Funding

## Competing interests

The authors declare no competing interests.

## Additional information

¹The Arctic University Museum of Norway, UiT - The Arctic University of Norway, NO-9037 Tromsø, Norway. ²Department of Evolutionary Biology, Ecology and Environmental Sciences, University of Barcelona, 08028 Barcelona, Spain. ³Institute of Marine Sciences (ICM), CSIC, 08003 Barcelona, Catalonia, Spain. ⁴Centre for Palaeogenetics, 10691 Stockholm, Sweden. ⁵Department of Geological Sciences, Stockholm University, 10691 Stockholm, Sweden. ⁶Department of Arctic and Marine Biology, UiT The Arctic University of Norway, Tromsø, Norway. ⁷Foundation Aubert, 1938 Champex-Lac, Switzerland. ⁸Department of

Plant Sciences, University of Geneva, 1292 Chambésy, Switzerland. [9]Institute of Plant Sciences and Oeschger Centre for Climate Change Research, University of Bern, 3012 Bern, Switzerland. [10]Department of Biosciences, University of Salzburg, 5020 Salzburg, Austria. [11]Department of Archaeology, University of York, YO1 7EP York, UK. [12]Chrono-Environnement, UMR 6249 CNRS, Université de Franche-Comté, Besançon, France. [13]Department of Environmental Sciences, University of Basel, 4056 Basel, Switzerland. [14]Université Grenoble-Alpes, Université Savoie Mont Blanc, CNRS, LECA, 38000 Grenoble, France. [15]EDYTEM, CNRS, Université Savoie Mont Blanc, 73000 Chambéry, France. [16]Department of Environmental Science and Policy, Università degli Studi di Milano, 20133 Milan, Italy. [17]Swiss Federal Research, Institute for Forest, Snow, and Landscape Research (WSL), 8903 Birmensdorf, Switzerland. [18]Department of Environmental System Science, Institute of Terrestrial Ecosystems, ETH Zurich, 8092 Zurich, Switzerland. [19]Department of Environment and Biodiversity, University of Salzburg, 5020 Salzburg, Austria. [20]Department of Botany, University of Innsbruck, 6020 Innsbruck, Austria. [21]Department of Geology, University of Innsbruck, 6020 Innsbruck, Austria. [22]Department of Ecology, University of Innsbruck, 6020 Innsbruck, Austria. [23]Faculty of Geographical and Geological Sciences, Adam Mickiewicz University, 61-680 Poznań, Poland. [24]Department of Chemistry, Biochemistry and Pharmaceutical Sciences & Oeschger Centre for Climate Change Research, University of Bern, 3012 Bern, Switzerland. ✉e-mail: sgarcespastor@gmail.com

## PhyloAlps Consortium

Sébastien Lavergne ⓘ [14] & Eric Coissac[14]

