## [Transparent Peer Review file · Nature Communications]

Wild and domesticated animal abundance is associated with greater late-Holocene alpine plant diversity

Corresponding Author: Dr Sandra Garcés-Pastor

Version 0:

Reviewer comments:

Reviewer #1

(Remarks to the Author)

The manuscript entitled "Wild and domesticate mammals drove late-Holocene alpine plant diversity" tackles an interesting question. It aims to unveiling the relationships between grazing of wild and domesticated mammals and vegetation from the Holocene. Overall, the manuscript is well written, scientifically sound, and of potential interest to researcher in various field. I have only some suggestions that might increase the clarity of the manuscript.

- 1) L98. The term "humans" used in conjunction with animals seems to suggest that the manuscript also analyses the impact of other human activities unrelated to pastoralism. Please consider changing the sentence to how climate, wild and farmed animals have shaped plant diversity, or something similar.
- 2) L98 The relationship with climate is worthy to be expanded. This could also help to better understand whether or how much the observed changes are due to animals themselves or to climate.
- 3) L127. Because the main topic is on the effect of animals on vegetation would be better to indicate not only the climatic preference of plants but also what kind of plants become more abundant (forbs, graminoids, shrubs or trees)
- 4) L149-150. Please indicate where the locations are located. In the Eastern, Central or Western Alps
- 5) L162. Please, indicate if you are considering only the presence or also the quantity of herbivores
- 6) L165-173. In this point would be worth to cite that the grazing is spatially structured, and this creates spatial heterogeneity of vegetation that in turn increases plant richness. See for example Adler, P., Raff, D. & Lauenroth, W. The effect of grazing on the spatial heterogeneity of vegetation. *Oecologia* 128, 465–479 (2001)
- 7) Please label the y-axis with the unit of measurement in Figure 2. Still in figure 2, it is not clear what is the black line. Is Hill N0 the subtitle of the figure 2A or the label of the black line? This is not very clear.
- 8) In Figure 3 and Table 1 the acronym Hill N0 is probably not very useful. In my opinion either you report in the figure caption for example "plant diversity measured as proportion of weighted PCR replicates" or you just indicate "Plant diversity" and the reader can see in M&M how you calculated this value

Reviewer #2

(Remarks to the Author)

I greatly enjoyed reading this manuscript. Scientifically, it is excellent, exciting, innovative, and rigorous. Unfortunately, the manuscript is slightly spoiled by the rather too many careless errors. I have listed those that I have spotted in the comments below.

Lines 15–43. Some of the addresses are not complete compared with others. Incomplete ones include line 19 and those without postcodes on lines 16–17, 20, 24–25, 26, 27, 35, 37, 39, and 41–42.

Line 9. I thought Fabian Rey (line 7) was now in Basel (address on line 29)

Line 48. Measure is a strong word; I suggest using estimate instead.

Line 57 and throughout. Use palaeoecology and palaeoecological to maintain consistency rather than mixing in paleoecology and paleoecological.

Line 77. Change poses to pose.

Line 90. Underrepresented may be better with a hyphen as in under-represented.

Line 125. Change Reed to Red.

Line 129. Change disappears to disappear.

Line 152. Add were after stock (i.e. stock were ...)

Line 159. Change fungi to fungal.
Line 175. Change forbs to forb.
Line 208. Change differs to differ.
Line 214. What is it?
Line 254. Better to write at than of (fieldwork at Swiss lakes).
Line 290. Better to write developed than performed.
Line 303. What does the * refer to?
Line 370. But Fig. S1 extends to 0 ka, not 4 ka.
Lines 380, 383. Palaeo-orography, rather than paleo orography.
Line 409. Extra closing bracket needed after UiT).
Lines 444, 471. Hill N0 really estimates richness.
Lines 471, 538. Hill N0, not Hill 0.
Line 529. Pollen richness from Giesecke et al. 2019 – where is this on Fig. 2?
Line 530. Change vegetal to vegetative.
Line 544. Change color to colour.
Line 604. Reference incomplete.
Lines 753–54. Is this complete?
Fig. S4. Should be Hill N0 and Hill N1, also in the labels for the y-axis in the figures.
Supplementary information Lake description. Page 9: Cyperaceae are not grasses. Page 10, line 196: change alpinum to alpinus.

Reviewer #3

(Remarks to the Author)
Review of Pastor et al.

This is an interesting paper that generates plant and animal sedaDNA signals across 14 alpine lakes to look at changing plant and animal diversity (wild and domesticates) as potential drivers or inhibitors of plant diversity.

Overall, this paper is well written and the message clear. I do have a few questions as to the data generation and a few other minor comments.

Lines 396 – 405.

The authors ran 17 negative and 18 positive controls.

What species did the authors detect as contamination? How was contamination eliminated or dealt with?

What positive controls were used and did the authors see the positive amplifications in their negative controls ever?

Some basic questions regarding the PCRs. Apologies, as some of the answers to these may be cited from previous literature, not read by the reviewer.

How long are the PCR amplicons for the trnL P6 and the 16s, generally?

What are the length variations between the different species detected and how do the GC contents vary? Are there any sequence variants in the primer binding sites for any of the detected taxa and have the authors tested their effects on preferential amplification?

If for example the authors place modern aliquots of the detected mammal DNA in equal ratios and amplify and sequence them, do the sequences represent a 1:1:1 and so on ratio?

The authors state that they carried out 8 PCR reactions for each sample (for each PCR reaction. The PCR reactions were library prepped and sequenced. It's unclear how many reads were generated per PCR reaction? Can the authors include that data somewhere?

Also, what was the duplication rates per reaction (how quickly did the sequencing runs saturate per library)? Attempting to get an idea on the level of detection (single molecules are multiple molecules starting the amplification).

What were the off-target amplifications if any ?

Was any qPCR performed to look at quantity of a certain species DNA in a lake (probe specific qPCR)? Figure 2 has a Y-axis for both plant and animal DNA that ranks to 100 – but it's unclear what that means? If for example all 8 replicates for one extraction at one lake generated Bos (8/8) – would that be 100 on the y-axis for that locale at that date?

For a taxa to be present the authors plant sequences from the PCR products had to have 100% identity to a known reference and represented by three reads per replicate and with a minimum of 10 total reads and three replicates across the entire data set.

To be clear, 8 PCR reactions and at least one read in three separate PCR reactions (3/8) to give the three read minimum for presence?

A minimum of 10 reads and three replicates across the entire data set- can the authors specify what this means? How are they determining reads number since PCR sequences will be clonal, one has no one of knowing if the reaction began from 1 or 100 molecules? 10 reads from 8 PCR reactions doesn't make sense, but I'm sure I'm missing something here.

Line 113 – “after removing potential contaminants’ – what were these and how were they first identified as contaminants and then simply removed.

Line 114 – ‘we retained 32 sequences representing...’ I count far less than 32 taxa in the lines below. What exactly are the 32 sequences?

Figure 2- can the authors include temperature reconstruction along the transect for comparison, since it's one of the variables mentioned as a possible driver of diversity.

Can the authors make a stronger case for the interaction of all factors? In the conclusion (lines 228) the authors claim it's a combination of multiple factors, but the main message appears to be strong evidence of Bos as drivers of diversity. I would argue that the climate portion of the forcing in the current draft is minimal if at all and would be beneficial to increase or at least balance with the conflicting pressures.

Overall, a very nice paper and I look forward to seeing it in print.

Version 1:

Reviewer comments:

Reviewer #1

(Remarks to the Author)

The authors improved the manuscript resolving all weaknesses underlined by reviewers. So, in my opinion the manuscript may be now published.

Reviewer #2

(Remarks to the Author)

I was pleased to read the revised version of this excellent manuscript. I am glad that the minor errors I pointed out in my first review have been addressed. However, on reading the revised version, I have noticed a few minor points.

Title - domesticated, not domesticate

l.47 Estimate the, not Estimaethe

l.51 - alpine - are the sites really sub-alpine

l.64 - 'the' before European

l.97 - niche construction, perhaps better to say niche development

l.165 - add 'the' before arrival

l.223 - shrub, not shrubs; tree, not trees

l.231 - insights, not insight

l.260 - Martin ? second name

l.310, 311 - OH twice ?correct

l.367 - Do you mean that the radiocarbon dates for the 14 sites and all the dates shown in Figure S1 are based on only 10-20 macrofossils per sequence? This implies that the AMS dating was done on one or two macrofossils per dated level. Is this correct?

l.391 - Palaeo-, not Paleo- and line 394

l.401, 436 - what is TMU?

l.447 - add (after PhylNorway

l.473,505,518,586,592,596 - Hill's N0 is a measure of richness rather than diversity which can be estimated by Hill's N1 or N2.

l.580 - vegetative, not vegetal

Figure S4 - Hill NO is a measure of richness, Hill N1 is a diversity measure

Figure S5 - richness, not diversity

Reviewer #3

(Remarks to the Author)

The authors have addressed the concerns I raised in PCR based analyses and some of the interpretations. I'm happy with the new manuscript which is more clear and a lovely study.

Dear reviewers,

Thank you very much for your valuable comments, which have significantly enhanced the quality of the manuscript.

We expanded on the relationship with climate and strengthened the discussion of the interaction between all factors in the conclusion. Additionally, we have clarified several parts of the Material and Methods, especially the sections: Statistical analysis, Sedimentary ancient DNA data generation, Database construction and *seadDNA* data analyses and Ecological indicators from plant *seadDNA*.

All reviewer suggestions have been addressed, leading to notable changes to the structure and content of the manuscript. Detailed responses to each comment are provided below.

Yours sincerely,

Sandra Garcés-Pastor and co-authors

Updated figures and tables:

Revised manuscript

Figure 2

Table 1

Supplementary Figure S10

Supplementary Data S9, S10, S11

Our detailed comments are given below.

REVIEWER COMMENTS

Reviewer #1 (Remarks to the Author):

The manuscript entitled “Wild and domesticated mammals drove late-Holocene alpine plant diversity” tackles an interesting question. It aims to unveiling the relationships between grazing of wild and domesticated mammals and vegetation from the Holocene. Overall, the manuscript is well written, scientifically sound, and of potential interest to researcher in various field. I have only some suggestions that might increase the clarity of the manuscript.

We thank the reviewer’s positive feedback and thoughtful, constructive comments. These insights have been invaluable in improving our manuscript and enhancing the clarity and depth of our study.

1) L98. The term "humans" used in conjunction with animals seems to suggest that the manuscript also analyses the impact of other human activities unrelated to pastoralism. Please consider changing the sentence to how climate, wild and farmed animals have shaped plant diversity, or something similar.

We changed the sentence to: “We applied *sedDNA* analysis to reveal how climate, along with wild and domesticated animals, has shaped plant diversity in different areas of the Alps.” Lines 101-103.

2) L98 The relationship with climate is worthy to be expanded. This could also help to better understand whether or how much the observed changes are due to animals themselves or to climate.

We have expanded the relationship with climate in the first paragraph of the introduction. “The relationship between climate and biodiversity in the Alps is complex, as climate affects factors such as temperature and precipitation, which in turn shape plant distribution and ecosystem dynamics. Short-term studies suggest that careful management through grazing may mitigate climate-change induced loss of species ³⁻⁵, but little is known about the long-term process. During the first half of the Holocene (~11 to 6 ka), climate was the main driver of vegetation. Rising temperatures initiated a successional shift, characterized by an initial increase in forbs, followed by a transition to shrubs and forests as warming progressed. Over the last 5 ka, human impact has had a greater impact on biodiversity of the Alps than climate change.” Lines 76-84.

3) L127. Because the main topic is on the effect of animals on vegetation, it would be better to indicate not only the climatic preference of plants but also what kind of plants become more abundant (forbs, graminoids, shrubs or trees)

We have added additional information on the most abundant taxa to enhance clarity. Lines 126-143.

4) L149-150. Please indicate where the locations are located. In the Eastern, Central or Western Alps

The sentence was adjusted: "In our lakes, donkeys are exclusively present in Grenouilles and Sangiatto (Western Alps), whilst camels are detected in one sample each from Grenouilles (0.7 ka) and Profond (0.4 ka)(Western Alps)." Lines 154-156.

5) L162. Please, indicate if you are considering only the presence or also the quantity of herbivores

We adapted the sentence: "To explore the relative influence of climate (temperature and precipitation) and herbivore abundance (wild and domesticates) on plant diversity, we employed Generalized Additive Mixed Models (GAM) (Table 1; see Methods)." Lines 169-171.

And also in the Material and Methods, Statistical analyses section: We employed multiple (generalised) additive mixed models (GAM) (68) to investigate past changes in plant diversity (Hill N0) in response to climate (temperature and precipitation), wild (ibex, chamois, deer) and domesticated (goat, sheep, cattle) mammal abundances. Lines 504-506.

6) L165-173. In this point, it would be worth to cite that grazing is spatially structured, and this creates spatial heterogeneity of vegetation that in turn increases plant richness. See for example Adler, P., Raff, D. & Lauenroth, W. The effect of grazing on the spatial heterogeneity of vegetation. *Oecologia* 128, 465–479 (2001).

We adapted the paragraph: "Wild herbivores may have shaped the ecosystem by grazing and trampling, creating different patches of vegetation that enhanced sunlight penetration to the ground and reducing competitive plant growth. Trampling facilitated litter incorporation into the soil, creating openings for new plant growth and, in turn, increasing plant diversity (23)." Lines 173-176.

7) Please label the y-axis with the unit of measurement in Figure 2. Still in figure 2, it is not clear what is the black line. Is Hill N0 the subtitle of the figure 2A or the label of the black line? This is not very clear.

We labelled the y-axis, improved the labelling of Subfigure 2A, and adjusted the pollen line to a broken style for better clarity.

8) In Figure 3 and Table 1 the acronym Hill N0 is probably not very useful. In my opinion either you report in the figure caption for example "plant diversity measured as proportion of weighted PCR replicates" or you just indicate "Plant diversity" and the reader can see in M&M how you calculated this value

We changed Hill N0 by plant diversity in Figure 3 and Table 1. However, we kept the Hill N0 in the labelling to show how the diversity was measured.

Reviewer #2 (Remarks to the Author):

I greatly enjoyed reading this manuscript. Scientifically, it is excellent, exciting, innovative, and rigorous. Unfortunately, the manuscript is slightly spoiled by the rather too many careless errors. I have listed those that I have spotted in the comments below.

We greatly appreciate the reviewer's positive remarks and constructive feedback. Their valuable insights and recommendations have contributed significantly to improving the quality and rigour of our manuscript.

Lines 15–43. Some of the addresses are not complete compared with others. Incomplete ones include line 19 and those without postcodes on lines 16–17, 20, 24–25, 26, 27, 35, 37, 39, and 41–42.

Addresses were completed

Line 9. I thought Fabian Rey (line 7) was now in Basel (address on line 29)

Affiliation was changed

Line 48. Measure is a strong word; I suggest using estimate instead.

The word was changed to "estimate".

Line 57 and throughout. Use palaeoecology and palaeoecological to maintain consistence rather than mixing in paleoecology and paleoecological.

The terms were addressed to palaeoecology (L56, L203, L309).

Line 77. Change poses to pose.

The word was changed to “pose”.

Line 90. Underrepresented may be better with a hyphen as in under-represented.

The hyphen was added.

Line 125. Change Reed to Red.

The word was changed to “Red”.

Line 129. Change disappears to disappear.

The word was changed to “disappear”.

Line 152. Add were after stock (i.e. stock were ...)

The word was added after stock.

Line 159. Change fungi to fungal.

The word was changed to “fungal”.

Line 175. Change forbs to forb.

The word was changed to “forb”.

Line 208. Change differs to differ.

The word was changed to “differ”.

Line 214. What is it?

We added “Furthermore, cattle...”

Line 254. Better to write at than of (fieldwork at Swiss lakes).

The word was changed to “at”.

Line 290. Better to write developed than performed.

The word was changed to “developed”.

Line 303. What does the * refer to?

The * marks author of correspondence, it was deleted in this part, as well as #.

Line 370. But Fig. S1 extends to 0 ka, not 4 ka.

Thanks for pointing this out, we have changed 4 to 0 ka.

Lines 380, 383. Palaeo-orography, rather than paleo orography.

We have added the hyphen.

Line 409. Extra closing bracket needed after UiT).

We have deleted the extra bracket.

Lines 444, 471. Hill N0 really estimates richness.

We appreciate the reviewer’s comment and agree that Hill N0 estimates richness. However, we chose to use the term "diversity" instead of "richness" to ensure clarity for a broader audience.

Lines 471, 538. Hill N0, not Hill 0.

The word “N0” was added.

Line 529. Pollen richness from Giesecke et al. 2019 – where is this on Fig. 2?

We changed Subfigure 2A to make the pollen richness from Giesecke et al. 2019 more visible.

Line 530. Change vegetal to vegetative.

The word was changed to “vegetative”.

Line 544. Change color to colour.

The word was changed to “colour”.

Line 604. Reference incomplete.

Reference was completed.

Lines 753–54. Is this complete?

The reference was addressed..

Fig. S4. Should be Hill N0 and Hill N1, also in the labels for the y-axis in the figures.

Done.

Supplementary information Lake description. Page 9: Cyperaceae are not grasses. Page 10, line 196: change alpinum to alpinus.

The sentence was rephrased, and the word changed to alpinus.

Reviewer #3 (Remarks to the Author):

Review of Pastor et al.

This is an interesting paper that generates plant and animal sedaDNA signals across 14 alpine lakes to look at changing plant and animal diversity (wild and domesticates) as potential drivers or inhibitors of plant diversity.

Overall, this paper is well written and the message clear. I do have a few questions as to the data generation and a few other minor comments.

Lines 396 – 405.

The authors ran 17 negative and 18 positive controls.

What species did the authors detect as contamination? How was contamination eliminated or dealt with?

Negative controls C1 to C6 were related to monitoring for contamination inside the clean lab. The species found in controls C1 and C2 were sampling-related, while C3 and C4 indicated extraction contamination. Controls C5 were associated with potential contamination when transferring DNA extracts from tubes to the plate for metabarcoding PCR setup, while C6 served as a negative PCR control. Later, controls C7 and C8 were post-PCR controls in the PCR lab, consisting of water and a synthetic standard. These are not true negative controls as we expect amplicon contamination in the post-PCR lab, but they allow us to keep an eye on the overall contamination in that room. As the sample PCR reactions are not exposed to the PCR-lab environment, only C1-C6 are informative for any contamination of concern to the study.

All species detected in the negative controls C1-C6 (see Supplementary Data S9 for plants and S10 for mammals) were considered contaminants. Consequently, we excluded these species from the analyses, plotting, and further interpretations. However, they remain in the plant sequence table (Supplementary Data S3) and are marked as contamination under the functional group column.

We added the following to the Material and Methods, Sedimentary ancient DNA data generation: “Several controls were used during the extraction and amplification process: controls C1 and C2 were related to sampling from the core, C3 and C4 to detect contamination during DNA extraction, and C5 were associated with potential contamination during the movement of DNA extracts from tubes to plates in preparation for metabarcoding PCR. At the same time, C6 served as a negative PCR control. C7 controls consist of DNA-free water added in the post-PCR laboratory and serves as our internal track of the amplicon contamination level in that room. The positive controls (C8) used in the post-PCR lab are synthetic standards (see Rijal et al. 2021), and their purpose is purely to check that the PCR has worked. As the sample PCR reactions are not exposed to the PCR-lab environment, C7 and C8 sequences were removed during the bioinformatic processing and not considered in downstream analyses.” Lines 405-414.

What positive controls were used and did the authors see the positive amplifications in their negative controls ever?

The information about positive controls has now been added (see above). Amplification in negative controls was occasionally observed and these are reported in Dataset Table 9.

Some basic questions regarding the PCRs. Apologies, as some of the answers to these may be cited from previous literature, not read by the reviewer.

How long are the PCR amplicons for the trnL P6 and the 16s, generally?

We have added the length information in the sedimentary ancient DNA data generation subsection of the material and methods: “For plants, the primer set targets the trnL P6 loop region of the chloroplast UAA intron (gh primers, 10-143 bp; ⁵²). These primers were specifically designed by Taberlet ⁵² to target two highly conserved catalytic regions of the trnL (UAA) intron. Although primer-binding sites mismatches are observed in some families such as Asteraceae (g: 0, h: 19.8%), Caryophyllaceae (g: 0.9%, h: 6.8%) and Crassulaceae (g: 0, h: 6.2%), the mismatch is $\leq 1\%$ for other families ⁵³. This locus has been shown to provide the highest taxonomic resolution in ancient plant sedaDNA studies ^{54,50,55,56}. For mammals, a section of the mitochondrial 16S locus was amplified (MamP007 primers, 60-84 bp; ⁵⁷)”. Lines 419-427.

The *trnL* P6 loop is a region of the chloroplast *trnL* (UAA) intron with a length of ca. 86 bp (10-143 bp, Taberlet et al. 2007). This locus has been highly effective in reconstructing plant communities in *sedaDNA* studies. Similarly, the mammalian mitochondrial 16S locus (amplified with MamP007 primers) has a length of 60–84 bp (Giguet-Covex et al. 2014). We used 8 bp unique primer tags for both the *trnL* P6 loop and the mammalian MamP007 primers to accurately assign sequence reads to the respective sample after the high-throughput sequencing.

- Taberlet, P. et al. Power and limitations of the chloroplast *trnL* (UAA) intron for plant DNA barcoding. *Nucleic Acids Res.* 35, e14 (2007).
- Giguet-Covex, C. et al. Long livestock farming history and human landscape shaping revealed by lake sediment DNA. *Nat. Commun.* 5, 3211 (2014).

What are the length variations between the different species detected and how do the GC contents vary?

We thank the reviewer for suggesting the inclusion of these summary statistics of our data set. The length of the amplified fragment is highly variable among species and ranges from 10 to 146 bp (Taberlet et al. 2007). A document with the average sequence length has been added to the Supplementary Data S11 Plant sample information (columns BY to CB, and at the bottom of the table).

We calculated the length and GC content for both the PhyloAlps p6 loop reference database and the ECOGEN Alps dataset, including all samples (controls and low-quality samples) as well as those containing at least 30 unique barcodes, which better represent high-quality samples (see table below). The observed average length was slightly lower compared to the reference database, while the GC content was slightly higher. This may be due to DNA preservation and/or bias. However, when examining the data on a per-lake basis, these metrics appeared relatively stable, with older samples showing similar average length and GC values to younger ones. This consistency suggests that the difference between the reference database and the lake data is ecological in nature. For instance, taxa present at higher elevations may have slightly shorter lengths and higher GC content.

Type	PhyloAlps (n = 4442)	All Alps samples (n = 75344)	Alps samples >= 30 barcodes (n = 73928)
Average length	50.83 bp (sd: 13.71)	46.99 bp (sd: 11.66)	47.02 bp (sd: 11.63)
Average GC	30.37% (sd: 7.98)	31% (sd: 7.2)	31% (sd: 7.19)
Average forward error	0.1425 (sd: 0.4642)	0.0859 (sd: 0.2924)	0.0845 (sd: 0.2903)
Average reverse error	0.4849 (sd: 0.6686)	0.4889 (sd: 0.6751)	0.4895 (sd: 0.6741)

We have added the following in the Material and Methods, Database construction and *sedDNA* data analysis: “The average length and GC content of our final sequence dataset were 47.02 ±11.63 bp and 31 ±7.19 %, respectively, which closely resemble the values from the total PhyloAlps P6loop reference database (50.83 ±13.71 bp and 30.37 ±7.98 %), thus confirming that the DNA is of high quality.” Lines 463-466.

Are there any sequence variants in the primer binding sites for any of the detected taxa and have the authors tested their effects on preferential amplification?

There are some sequence variants in the primer binding sites and so we have added this limitation to the Materials and Methods, Sedimentary ancient DNA data generation: “Although primer-binding sites mismatches are observed in some families such as Asteraceae (g: 0, h: 19.8%), Caryophyllaceae (g: 0.9%, h: 6.8%) and Crassulaceae (g: 0, h: 6.2%), the mismatch is ≤ 1% for other families ⁵³.” on Lines 422-424.

We have not explicitly tested their effects on preferential amplification, but note that primer bias is a well appreciated limitation of the metabarcoding approach.

If for example the authors place modern aliquots of the detected mammal DNA in equal ratios and amplify and sequence them, do the sequences represent a 1:1:1 and so on ratio?

No, unfortunately the ratio is unlikely to be 1:1. This is due to a myriad of biases, including the primer bias listed above. Other biases include that the variation in abundance of template DNA an individual animal will shed in a catchment will vary with a lot of factors. These include time spent there, defecation or not, moulting, or potentially even dying close to the lake. The overall abundance of template DNA from mammals is much lower than that of plants, and

therefore they are less reliably detected. Also, as only a few mammals are detected in each sample, the % of reads are unreliable due to stochasticity. While we assume that, as with plants, DNA detection increases with DNA template abundance, we approach using this quantitatively with caution.

The authors state that they carried out 8 PCR reactions for each sample (for each PCR reaction). The PCR reactions were library prepped and sequenced. It's unclear how many reads were generated per PCR reaction? Can the authors include that data somewhere?

A document with the "raw" reads for each PCR after demultiplexing step has been added to Supplementary Data S11 (Plant sample information, columns B to I).

Also, what was the duplication rates per reaction (how quickly did the sequencing runs saturate per library)? Attempting to get an idea on the level of detection (single molecules are multiple molecules starting the amplification).

Unfortunately, this cannot be determined with the metabarcoding approach, as we do not use qPCR for single-species quantification or use the Unique Molecular Identifiers (UMIs) that would be required to separate PCR duplicates from multiple original template molecules ('biological duplicates', if you will).

What were the off-target amplifications if any ?

We added this to Material and Methods, Database construction and *sedDNA* data analysis: "Off-target amplifications for g-h primer (bryophytes and algae) are reported in Supplementary Data S3 and MamP007 (birds, fish, and invertebrates) in Supplementary Data S4." Lines 461-463.

Was any qPCR performed to look at quantity of a certain species DNA in a lake (probe specific qPCR)?

Our study focuses on community-scale metabarcoding rather than single-species detection using qPCR. The qPCR was performed solely to quantify the library before sequencing. Using end-point qPCR is a valuable approach that would be interesting to test, but one would need to design species-specific primers and run separate qPCR assays per species.

Figure 2 has a Y-axis for both plant and animal DNA that ranks to 100 – but it's unclear what that means? If for example all 8 replicates for one extraction at one lake generated Bos (8/8) – would that be 100 on the y-axis for that locale at that date?

We added “Relative abundance of *sedaDNA*” to the y-axis.

We also added a short explanation in the Material and Methods, Ecological indicators from plant *sedaDNA*: The RAI it is a proportional index for each taxon within a sample, calculated by multiplying the proportion of obtained reads by the proportion of weighted PCR replicates. The latter serves as a barcode relative detectability measure to account for variations in the relative read counts across all retained barcodes within a sample ^{2,51}. These values are then normalized for plotting (scaled), dividing by the sum of all RAIs for all species plotted in the figure”. Lines 497-502.

We also have added this sentence to Figure 2 label: “all plant and mammal values are presented as RAI scaled according to the plotted groups.”

- Garcés-Pastor, S. et al. High resolution ancient sedimentary DNA shows that alpine plant diversity is associated with human land use and climate change. *Nat. Commun.* 13, 6559 (2022).
- Rijal, D. P. et al. Sedimentary ancient DNA shows terrestrial plant richness continuously increased over the Holocene in northern Fennoscandia. *Sci. Adv.* 7, eabf9557 (2021).

For a taxa to be present the authors plant sequences from the PCR products had to have 100% identity to a known reference and represented by three reads per replicate and with a minimum of 10 total reads and three replicates across the entire data set.

To be clear, 8 PCR reactions and at least one read in three separate PCR reactions (3/8) to give the three read minimum for presence?

Our apologies for the confusion. We required three reads to be present in a PCR replicate (ie. a sequenced PCR product) in order to call a detection within that replicate. So in the example the reviewer gives, with one read per replicate in 3/8 replicates, there would be zero detections and so 0/8 replicates.

We have updated “three reads per replicate” to “three reads in a PCR replicate” to clarify on Lines 450-452.

A minimum of 10 reads and three replicates across the entire data set- can the authors specify what this means? How are they determining reads number since PCR sequences will be clonal, one has no one of knowing if the reaction began from 1 or 100 molecules? 10 reads from 8 PCR reactions doesn't make sense, but I'm sure I'm missing something here.

To clarify, for these thresholds, this is across the entire dataset (ie. all samples). So if taxon A, for example, was only detected in two PCR reactions across the whole dataset (not just within a sample, as the reviewer implies), then taxon A is filtered out. Our choice of these detection thresholds is because there is always a trade-off between keeping true positives and getting rid of false positives (Ficetola et al. 2015). In Alsos et al. 2018 (methods + Supplementary Table S3), we used vegetation surveys as documentation of true positives, whereas species never observed in the region were assumed to represent false positives. We then tested different combinations of filtering criteria using minimum numbers of reads, minimum numbers of PCR repeats as well as occurrence in negative controls compared to samples. The optimal combination was selected based on the optimal ratio between true positive kept and false positives lost. Based on these results, we have set our default bioinformatic pipeline to 10 reads and 3 replicates. After this filter has been applied, we still do a manual check that the data make sense in terms of biogeography of the taxa identified.

We have now made this clearer by referring to this study for the cut-off level chosen: "Only plant sequences with a 100% match to a reference sequence, represented by three reads in a PCR replicate, and with a minimum of 10 total reads and three replicates across the entire data set were retained ⁶⁴." Lines 450-452.

- Ficetola, G. F. et al. Replication levels, false presences and the estimation of the presence/absence from eDNA metabarcoding data. *Mol. Ecol. Resour* 15, 543–556 (2015).
- Alsos, I. G. et al. Plant DNA metabarcoding of lake sediments: how does it represent the contemporary vegetation. *PLoS ONE* 13, e0195403 (2018).

Line 113 – "after removing potential contaminants' – what were these and how were they first identified as contaminants and then simply removed.

We have added this information into the Material and Methods section of Database construction and *sedDNA* data analysis: "We observed occasional contamination of cattle (1%), sheep (5%), and deer (3%) in our 100 negative controls (data S10). Consequently, we need to approach sporadic appearances with caution. Although we retained detections from all samples,

we interpret sporadic occurrences of these taxa, defined as single, isolated PCR replicate detections, as possible deriving from contamination. We note that these occur in otherwise low diversity samples that are most at risk of sporadic contamination detections (e.g., sheep, goat and cattle before 8 ka) (Fig. 3 and Supplementary Fig. 9). Thus, for this study, we excluded single occurrences of domestic cattle (*Bos taurus*), sheep (*Ovis aries*), and goat (*Capra hircus*) before 8 ka, such as those in Bretaye, Sulsseewli, Sulzskar, Colbricon, Emines and Poursollet (Supplementary Data S7).“ Lines 475-485.

We have also listed all plant and mammal contaminant detections in Supplementary Data S9 and S10.

Line 114 – ‘we retained 32 sequences representing...’ I count far less than 32 taxa in the lines below. What exactly are the 32 sequences?

We thank the reviewer for pointing this out. Some species have more than one sequence (or haplotype) and so, if assigned to the same taxon, we collapsed these to that taxon. There are therefore fewer taxa than there are retained sequences. We have added the word "variants" to the sentence (Lines 117-119) and included a reference to the dataset to clarify the statement. We also note that this is described in the Material and Methods, Database construction and *sedDNA* data analysis section, on Lines 475-485.

Figure 2- can the authors include temperature reconstruction along the transect for comparison, since it's one of the variables mentioned as a possible driver of diversity.

To explore temperature as a driver of diversity, we extracted temperature reconstructions from the geographic coordinates of each lake. Including 14 individual temperature lines for each lake or a mean value in Figure 2 could lead to misinterpretation. To address this, we added a Chironomid-reconstructed temperature line in Figure 2. We also provided a new Supplementary Figure S10, illustrating the temperature and precipitation data for all 14 lakes over the past 10 ka.

Can the authors make a stronger case for the interaction of all factors? In the conclusion (lines 228) the authors claim it's a combination of multiple factors, but the main message appears to be strong evidence of *Bos* as drivers of diversity. I would argue that the climate portion of the forcing in the current draft is minimal if at all and would be beneficial to increase or at least balance with the conflicting pressures.

We have added some sentences to better integrate the relationship between climate, herbivores, and climate change. “Regarding climate, precipitation had a minor impact on total

plant diversity, while temperature exhibited a negative correlation. However, to mitigate the effects of rising temperatures, the influence of herbivores becomes crucial.” Lines 241-243.

Overall, a very nice paper and I look forward to seeing it in print.

We thank the reviewer for their kind and constructive comments, and share the same sentiment.

Dear editor and reviewers,

Thank you very much for your valuable comments, which have greatly improved the quality of our manuscript.

We have addressed Reviewer 2's comments regarding misspelt words, radiocarbon dates, and especially Hill's N0 as a measure of richness.

Additionally, we have implemented all the changes recommended in the checklist.

All reviewer suggestions have been addressed, leading to notable changes to the structure and content of the manuscript. Detailed responses to each comment are provided below.

Yours sincerely,

Sandra Garcés-Pastor and co-authors

Our detailed comments are given below.

REVIEWER COMMENTS

Reviewer #1 (Remarks to the Author):

The authors improved the manuscript resolving all weaknesses underlined by reviewers. So, in my opinion the manuscript may be now published.

We thank the reviewer's positive feedback and thoughtful constructive comments.

Reviewer #2 (Remarks to the Author):

I was pleased to read the revised version of this excellent manuscript. I am glad that the minor errors I pointed out in my first review have been addressed. However, on reading the revised version, I have noticed a few minor points.

We thank the reviewer for the nice words and detailed revision. We have corrected the minor points spotted.

Title - domesticated, not domesticate

Corrected

I.47 Estimate the, not Estimaethe

Corrected

I.51 - alpine - are the sites really sub-alpine

Corrected

I.64 - 'the' before European

Corrected

I.97 - niche construction, perhaps better to say niche development

The word was changed to "development".

I.165 - add 'the' before arrival

Corrected

I.223 - shrub, not shrubs; tree, not trees

Corrected

I.231 - insights, not insight

Corrected

I.260 - Martin ? second name

Corrected

I.310, 311 - OH twice ?correct

Corrected

I.367 - Do you mean that the radiocarbon dates for the 14 sites and all the dates shown in Figure S1 are based on only 10-20 macrofossils per sequence? This implies that the AMS dating was done on one or two macrofossils per dated level. Is this correct?

We thank the reviewer for this, to clarify the text, we added the sentence “For each dated level, AMS dating was typically performed on one or two well-preserved macrofossils.”

I.391 - Palaeo-, not Paleo- and line 394

Corrected

I.401, 436 - what is TMU?

TMU is the Arctic University Museum of Norway in Tromsø (TMU, Norway), as defined in the "Study Sites and core sampling" section of the Methods.

I.447 - add (after PhyloNorway

Corrected

I.473,505,518,586,592,596 - Hill's N0 is a measure of richness rather than diversity which can be estimated by Hill's N1 or N2.

We appreciate the reviewer’s comment and agree that Hill N0 strictly estimates richness. However, we chose to use the term "diversity" instead of "richness" to ensure clarity for a broader audience. Diversity is a broad term, and generally also covers the number of species (richness = Hill N0). A reader familiar with the Hill numbers will know that Hill N0 refers to the number of taxa without taking abundance into account.

To avoid any confusion, we added this at the first time we use “diversity” referring to Hill N0: “(commonly referred to as species richness)” (Lines 419-420)

I.580 - vegetative, not vegetal

Corrected

Figure S4 - Hill N0 is a measure of richness, Hill N1 is a diversity measure

We have added the term "richness" after Hill N0.

Figure S5 - richness, not diversity

We added the term "richness" to the figure caption: “Total plant species diversity and diversity for each growth form group measured as Hill N0 (richness). Smoothed curves using Span=0.25.”

Reviewer #3 (Remarks to the Author):

The authors have addressed the concerns I raised in PCR based analyses and some of the interpretations. I'm happy with the new manuscript which is more clear and a lovely study.

We thank the reviewer for their thoughtful and constructive comments.